

# Multiplet supercurrent in Josephson tunneling circuits

**André Melo**[1]$^\star$, **Valla Fatemi**[2] **and Anton R. Akhmerov**[1]

**1** Kavli Institute of Nanoscience, Delft University of Technology,
P.O. Box 4056, 2600 GA Delft, The Netherlands
**2** Department of Applied Physics, Yale University, New Haven, CT 06520, USA

$\star$ am@andremelo.org
See also: online presentation recording.

## Abstract

The multi-terminal Josephson effect allows DC supercurrent to flow at finite commensurate voltages. Existing proposals to realize this effect rely on nonlocal Andreev processes in superconductor-normal-superconductor junctions. However, this approach requires precise control over microscopic states and is obscured by dissipative current. We show that standard tunnel Josephson circuits also support multiplet supercurrent mediated only by local tunneling processes. Furthermore, we observe that the supercurrents persist even in the high charging energy regime in which only sequential Cooper transfers are allowed. Finally, we demonstrate that the multiplet supercurrent in these circuits has a quantum geometric component that is distinguishable from the well-known adiabatic contribution.

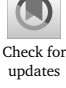

The DC Josephson effect allows coherent transport of Cooper pairs across two-terminal superconducting junctions at zero voltage [1]. At finite DC voltages the phase difference across the junction advances linearly in time, resulting in a pure AC supercurrent. A dissipative DC current may also arise due to multiple Andreev reflections [2]. However, charge transfers across terminals of a voltage-biased junction cost energy and thus no DC supercurrent can flow.

Junctions with additional terminals biased at commensurate voltages support energy-conserving processes that transfer charge between multiple electrodes. The simplest setup where this can occur is a three-terminal junction, where two voltage-biased terminals each transfer $n_1$ and $n_2$ Cooper pairs to a grounded terminal. At commensurate voltages $2en_1V_1 = -2en_2V_2$ this is a coherent and energy-conserving process that allows DC supercurrent. Several experimental works reported increased conductance at commensurate voltages as a signature of multiplet supercurrent in Josephson elements with weak links made of diffusive normal metals [3], InAs nanowires [4], and graphene [5].

So far, theoretical interpretations of these experiments rely on Andreev physics associated with highly transparent superconductor-normal-superconductor (SNS) junctions. In particular, the normal region must host nonlocal Andreev states that extend to multiple terminals and

mediate transport of charge through nonlocal Andreev processes [6–9] (see Fig. 1(a)). This mechanism is nontrivial because it is not guaranteed that a single state propagates to all three junctions, which may imply that multiplet supercurrent is a fragile phenomenon requiring fine tuning of microscopic aspects of the normal scattering region.

One may ask if this delicate microscopic process is the only mechanism that admits multiplet supercurrent. We draw inspiration from a problem in a similar context: multi-terminal SNS Josephson junctions were proposed as a platform for non-trivial band topology, where the superconducting phases play the role of crystal momenta [10]. Recent works showed that tunnel Josephson junction circuits are capable of encoding the same physics in collective electronic modes, rather than the fermionic degrees of freedom in the multi-terminal weak link [11, 12].

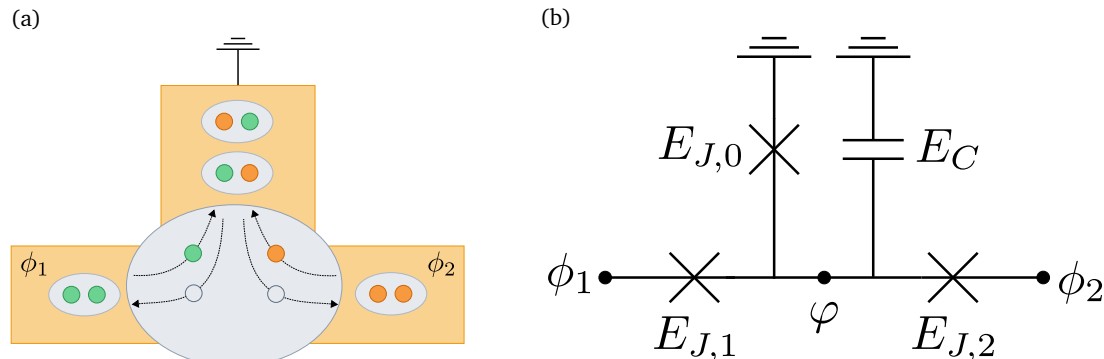

Figure 1: Superconducting devices that carry multiplet supercurrent at commensurate DC voltages. (a) A high transparency three-terminal Josephson junction supports multiplet supercurrent through non-local Andreev processes. When voltages applied to terminals 1 and 2 satisfy $V_1 = -V_2$, each biased terminal may transfer one Cooper pair to the grounded terminal through crossed Andreev reflections, resulting in quartet supercurrent. (b) A Josephson tunneling circuit also supports multiplet supercurrents. Even when the central island has a large charging energy, multiplet supercurrent still flows despite being carried only by single Cooper pair transfers.

In this work, we show that voltage-biased circuits of Josephson tunnel junctions also generate multiplet supercurrent, and we elucidate two types of contributions: an adiabatic component and a quantum geometric component. In contrast with its SNS counterpart, these circuits mediate the transport of multiplets through the collective behavior of the superconducting circuit, rather than microscopic multi-terminal Andreev processes. Furthermore, our proposal is experimentally tractable because tunnel junctions are standard building blocks of experimental superconducting devices.

We begin by analyzing the minimal tunneling circuit in Fig. 1(b) in the zero charging energy limit, $E_C = 0$, which may be referred to as the classical limit of the circuit. We assume that damping in the circuit allows treating the evolution of $\varphi$ adiabatically.

The circuit energy as a function of the superconducting phases is $E(\varphi, \phi_1, \phi_2) = -E_{J,0} \cos(\varphi) - E_{J,1} \cos(\phi_1 - \varphi) - E_{J,2} \cos(\phi_1 - \varphi)$, where the phases of the voltage-biased terminals evolve as $\dot{\phi}_i = V_i / \Phi_0$, where $\Phi_0 = \hbar/2e$ is the reduced magnetic flux quantum. Minimizing the circuit energy $E$ with respect to $\varphi$ for fixed $(\phi_1, \phi_2)$ gives the condition

$$\tan(\varphi) = \frac{E_{J,1} \sin(\phi_1) + E_{J,2} \sin(\phi_2)}{E_{J,0} + E_{J,1} \cos(\phi_1) + E_{J,2} \cos(\phi_2)} \ . \tag{1}$$

We then obtain the supercurrent flowing to ground using the Josephson relation

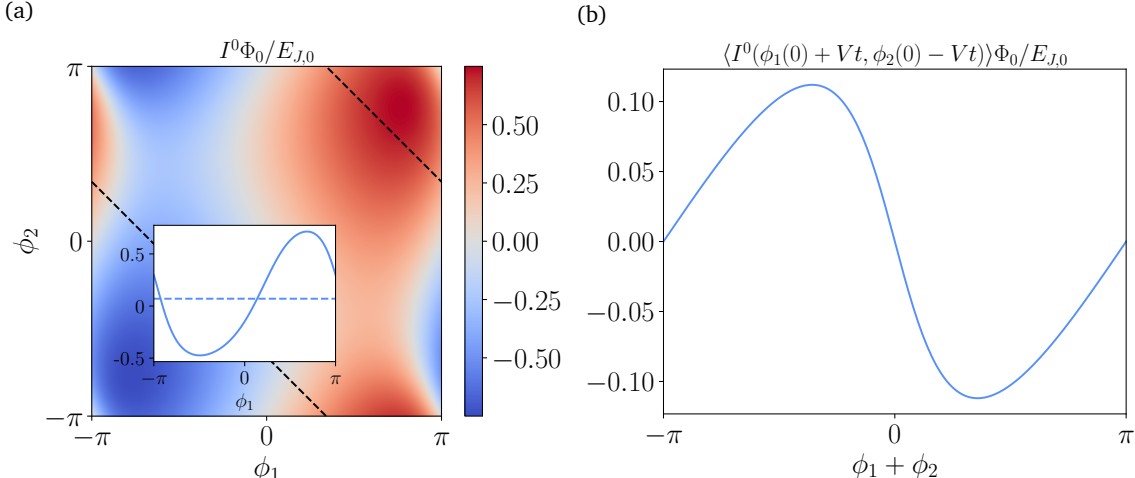

Figure 2: Currents across the $E_{J,0}$ junction in the circuit of Fig. 1 with $E_{J,1} = E_{J,0}/2, E_{J,2} = E_{J,0}/4, E_C = 0$. (a) Instantaneous current at fixed $\phi_1, \phi_2$. The inset shows the current along the quartet line $\phi_1 + \phi_2 = -1$ (black dashed line). Since the average (blue dashed line) is finite there is a quartet DC supercurrent. (b) Average current along quartet trajectories with different phase offsets.

$I^0(\phi_1, \phi_2) = E_{J,0} \sin(\varphi)/\Phi_0$. In Fig. 2(a) we plot $I^0(\phi_1, \phi_2)$ for a circuit with $E_{J,1} = E_{J,0}/2$ and $E_{J,2} = E_{J,0}/4$. Because the supercurrent is a periodic function of both $\phi_1$ and $\phi_2$ it admits a Fourier expansion

$$I^0(\phi_1, \phi_2) = \sum_{n,m} I^0_{nm} \sin(n\phi_1 + m\phi_2). \tag{2}$$

The $n, m$ harmonic in Eq. (2) is associated with transfering $n$ $(m)$ Cooper pairs from terminal 1 (2) and $n+m$ to the reference terminal [8]. If terminals 1 and 2 are biased with commensurate DC voltages $nV_1 + mV_2 = 0$, then harmonics with the ratio $n/m$ become resonant. Thus, a net DC current is produced if any of those Fourier components are nonzero, $I^0_{nm} \neq 0$. In the following we focus on the quartet supercurrent appearing when $V_1 + V_2 = 0$, i.e. when $n = m = 1$; however, calculations for higher harmonics are analogous. To check whether the circuit supports quartet supercurrent we plot the average current $\langle I^0(\phi_1(0) + Vt, \phi_2(0) - Vt) \rangle$ as a function of the phase offset in Fig. 2(b). We observe that the average is finite as long as the phase offset $\phi_1 + \phi_2$ (mod $2\pi$) $\notin \{0, \pi\}$, confirming that the circuit carries quartet supercurrent proportional to the critical current of the junction array.

We now investigate the role of quantum fluctuations in the circuit by including the charging energy of the superconducting island. The circuit Hamiltonian then reads

$$H = E_C(\hat{n} - n_g)^2 - E_{J,0}\cos(\hat{\varphi}) - E_{J,1}\cos(\hat{\varphi} - \phi_1) - E_{J,2}\cos(\hat{\varphi} - \phi_2), \tag{3}$$

where $\hat{n}$ is the number of Cooper pairs in the island, $n_g$ is the island offset charge, and $\hat{\varphi}$ is now promoted to a Hermitian operator conjugate to $\hat{n}$. In the adiabatic approximation in which the bias voltages are small enough to prevent Landau-Zener transition [13], the current flowing to ground equals

$$I^0_{\text{adiab.}} = \frac{1}{\Phi_0}\left(\frac{\partial E}{\partial \phi_1} + \frac{\partial E}{\partial \phi_2}\right), \tag{4}$$

where $E$ is the energy of the ground state. In Fig. 3(a) we show the resulting current in a circuit with $E_C = 30E_{J,0}$ and $n_g = 0$. We observe a similar functional dependence to that of the classical supercurrent in Fig. 1(a), indicating that the quartet supercurrent persists in the

presence of large charge fluctuations. At the same time, the magnitude of the supercurrent is significantly smaller than in the classical limit. In order to more systematically determine the effect of a large charging energy on the magnitude of supercurrent, we analytically compute $I^0(\phi_1, \phi_2)$ in the high charging energy limit. Near the charge degeneracy point $n_g = 0.5$ the system's dynamics are restricted to the two lowest charge states $|0\rangle$ and $|1\rangle$. The low-lying spectrum is then well approximated by the effective two-level Hamiltonian

$$H = \begin{bmatrix} 0 & \frac{1}{2}\left(E_{J,0} + E_{J,1}e^{i\phi_1} + E_{J,2}e^{i\phi_2}\right) \\ \frac{1}{2}\left(E_{J,0} + E_{J,1}e^{-i\phi_1} + E_{J,2}e^{-i\phi_2}\right) & E_1 \end{bmatrix}, \tag{5}$$

where we set the energy of $|0\rangle$ to zero and $E_1 = E_C(1 - 2n_g)$ is the energy of $|1\rangle$. The ground state energy reads

$$E = \frac{1}{2}\left(E_1 - \sqrt{E_1^2 + |E_{J,0} + E_{J,1}e^{i\phi_1} + E_{J,2}e^{i\phi_2}|^2}\right). \tag{6}$$

Using Eq. (4) we obtain the supercurrent flowing to ground:

$$I^0 = \frac{E_{J,0}(E_{J,1}\sin\phi_1 + E_{J,2}\sin\phi_2)}{2\sqrt{E_1^2 + |E_{J,0} + E_{J,1}e^{i\phi_1} + E_{J,2}e^{i\phi_2}|^2}}. \tag{7}$$

When $n_g \neq 0.5$, the charge degeneracy is broken ($E_1 \neq 0$). This supresses charge transfers to the island and thus the supercurrent vanishes as $E_C \rightarrow \infty$ (blue line in Fig. 3(b)). However, at the charge degeneracy point $E_1 = 0$ the supercurrent $I^0(\phi_1, \phi_2)$ becomes independent of $E_C$ (orange line in Fig. 3(b)). Remarkably, this implies that the quartet supercurrent is carried only by sequential Cooper pair transfers.

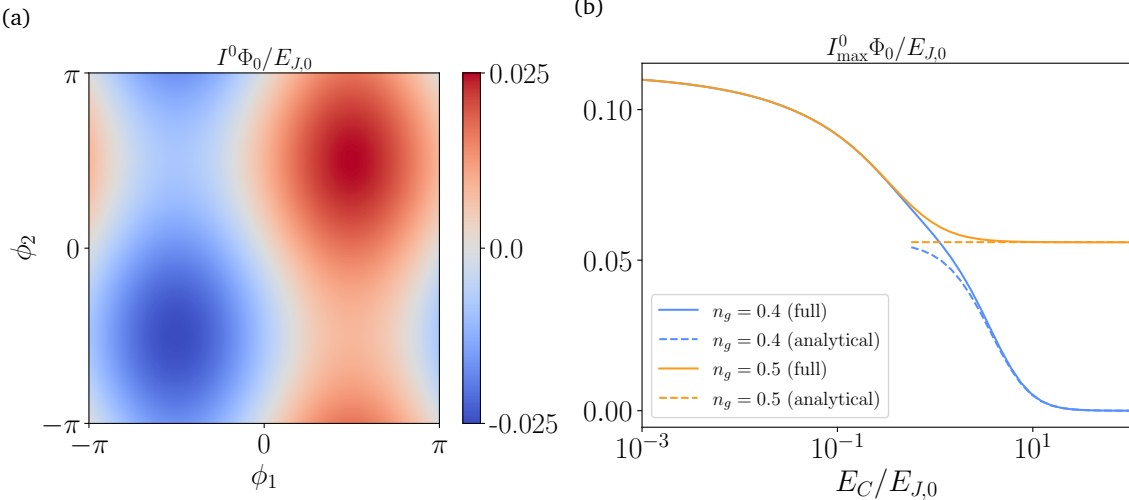

Figure 3: (a) Instantaneous current across the $E_{J,0}$ junction in the circuit of Fig. 1(a) with $E_{J,1} = E_{J,0}/2, E_{J,2} = E_{J,0}/4, E_C = 30E_{J,0}, n_g = 0$. (b) critical quartet supercurrent: the current $I_0$ maximized over the average phase of the electrodes $\max_{\phi_1+\phi_2}\langle I^0(\phi_1(0) + Vt, \phi_2(0) - Vt)\rangle$. When $n_g = 1/2$ charge states $|0\rangle$ and $|1\rangle$ are degenerate, allowing quartet supercurrent to flow with single Cooper pair transfers. The solid lines are obtained by numerically diagonalizing the full Hamiltonian (3), while the dashed lines are given by the approximate analytical expression (7), valid in the high $E_C$ limit.

When the bias voltages are commensurate, the closed trajectory in the $\phi_1, \phi_2$ parameter space results in the accumulation of a Berry phase with each cycle. While $\phi_1, \phi_2$ vary with

time, the instantaneous geometric contribution to the current is [10, 14, 15]

$$I^0_{\text{Berry}} = -2e\left(\Omega_{12}\dot{\phi}_1 + \Omega_{21}\dot{\phi}_2\right), \tag{8}$$

where the Berry curvature of the ground state $|\psi\rangle$ is given by

$$\Omega_{\alpha\beta} = -2\text{Im}\left\langle \frac{\partial\psi}{\partial\phi_\alpha} \bigg| \frac{\partial\psi}{\partial\phi_\beta} \right\rangle. \tag{9}$$

The quartet supercurrent then arises from the average of $I^0_{\text{Berry}}$ along a trajectory in phase space satisfying $\dot{\phi}_1 = -\dot{\phi}_2 = V/\Phi_0$, for which we simplify the instantaneous geometric current to

$$I^0_{\text{Berry}} = -\frac{(4e)^2}{h}\pi\Omega_{12}V, \tag{10}$$

where we used the relation $\Omega_{12} = -\Omega_{21}$. In contrast with the adiabatic term of Eq. (4), this current scales proportionally with the applied voltage. This allows the possibility of separately identifying the adiabatic and geometric parts.

The geometric quartet supercurrent requires additional conditions on the circuit's parameters. At charge-inversion invariant points $n_g \in \{0, 1/2\}$, the Hamiltonian (3) is both time-reversal and charge-inversion symmetric [11] and hence the Berry curvature vanishes. Away from these points the Berry curvature becomes finite; however, if $E_{J,1} = E_{J,2}$ it is antisymmetric along the quartet trajectories, i.e. $\Omega(\phi_1, \phi_2) = -\Omega(\phi_2, \phi_1)$. As a result, the average Berry curvature along a quartet trajectory $(\phi_1(0) + Vt, \phi_2(0) - Vt)$ vanishes regardless of the offset phase $\phi_1 + \phi_2$. When the Josephson energies differ (i.e. $E_{J,1} \neq E_{J,2}$), the Berry curvature landscape 'shears', resulting in a finite average on a quartet trajectory. As an example, in Fig. 4 we show the Berry curvature of a circuit with $E_C = E_{J_0}$ and $n_g = 0.7$. We observe that the average $\langle\Omega_{12}(\phi_1(0) + Vt, \phi_2(0) - Vt)\rangle$ is finite provided that $\phi_1 + \phi_2 \pmod{2\pi} \notin \{0, \pi\}$, resulting in the quantum geometric contribution to the quartet supercurrent.

Our proposal to produce multiplet supercurrent has two main advantages over its existing SNS counterpart. First, SNS devices require tuning wave functions of Andreev bound states that depend strongly on the microscopic details of the junction. Additionally, fabricating multi-terminal junctions is nontrivial [16]. In contrast, fabricating tunneling circuits with designed parameters (charging and Josephson energies) is a relatively routine procedure. Finally, SNS junctions have significantly larger dissipation due to the low resistance of the normal region [17].

Turning to existing experimental work [3–5, 16], we note that the most qualitative signatures of SNS-based multiplet supercurrents are, to the best of our knowledge, indistinguishable from those of tunnel-based junctions. Furthermore, it is known that the conventional Cooper pair transisor in the deep charging regime ($E_C \gg E_J$) has the same Hamiltonian as a single-level quantum dot with weak coupling strengths $\Gamma \ll \Delta$ to a pair of superconducting reservoirs [18]. In this analogy between the SNS and SIS devices, $\Gamma$ relates to $E_J$, and a level offset energy relates to offset charge. The same analogy extends to the multi-terminal case [19]. Such dots would exhibit the same kind of multiplet supercurrent described in our work. On the other hand, multiplet processes that entangle Cooper pairs from different leads require intermediate states with broken Cooper pairs, and thus they would be suppressed by factors of $\Gamma/\Delta$. Many of the Andreev levels of experimental multiterminal and multichannel SNS devices may be weakly coupled to the superconducting reservoirs, and those levels would predominantly contribute the kind of multiplet supercurrent described here. Thus, our results suggest that the multiplet supercurrent observed in SNS devices may have an alternative contribution arising from local Cooper pair transfers.

Moving forward, a question relevant for experimental implementation is how this device performs in a realistic environment including load circuit and environmental noise [20]. Another interesting avenue of further investigation would be to design tunneling circuits that

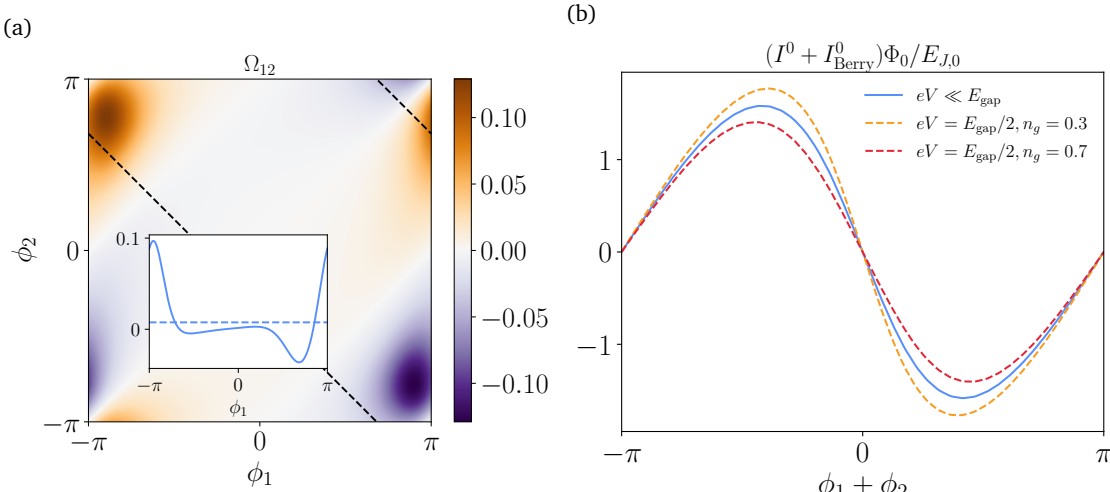

Figure 4: (a) Berry curvature of the ground state of the circuit in Fig. 1(b) with $E_{J,1} = E_{J,0}/2, E_{J,2} = E_{J,0}/4, E_C = E_{J,0}$ and $n_g = 0.7$. The inset shows a cut along $\phi_1 + \phi_2 = -1$ (black dashed line). Because the average (blue dashed line) is finite, the quartet supercurrent has a geometric contribution. (b) Quartet current-phase relation of the circuit in (a) with geometric corrections. The scale of the voltage is set by the minimum spectral gap $E_{\text{gap}}$ over the Brillouin zone. Reflecting $n_g$ about 0.5 flips the sign of the Berry curvature and hence of the geometric component of the supercurrent.

allow coherent control of the collective motion of quartets. Such a device could serve as a building block for parity-protetected $\cos 2\varphi$ electromagnetic qubits [21–23].

## Acknowledgements

The authors acknowledge useful discussions with Radoica Draškić, and V.F. acknowledges useful discussions with Nicholas Frattini and Pavel D. Kurilovich.

**Data availability**  The code used to generate the figures is available on Zenodo [24].

**Author contributions**  A.A. formulated the initial idea and supervised the project together with V.F. A.M. identified the role of the adiabatic multiplet supercurrent and performed the numerical and analytical calculations with input from A.A. and V.F. All authors contributed to writing the manuscript.

**Funding information**  This work was supported by the Netherlands Organization for Scientific Research (NWO/OCW), as part of the Frontiers of Nanoscience program and an NWO VIDI grant 016.Vidi.189.180.

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
