# Peer review of "Multiplet supercurrent in Josephson tunneling circuits"

_SciPost Physics, doi:SciPost Phys. 12, 017 (2022)_

## Round 3 · Referee Report · Regis Melin (Referee 1) · 2021-7-22

Report

1) When reading the abstract and the manuscript, I did not understand whether or not the authors want to deliver the message that Coulomb interactions are at work in the three recent experiments reporting the quartets. At present time, the message of the manuscript is ambiguous in this respect and the paper could be interpreted as bringing discredit on previous approaches based on Andreev processes, on the basis of claims that are not scientifically demonstrated in the paper. If the authors really want to make those claims, they should work out their model in the light of a thorough comparison with the available set of experimental data. They should evaluate all of the relevant quantities (current and current-current cross-correlations) that were measured in the experiments and determine how they vary as a function of the bias voltage. They should also address within their approach the experimentally relevant large or intermediate values of the interface transparencies. But maybe, this is just a matter of communication and choice of the words. Nevertheless, I ask the authors to clarify in the revised version that their new proposal does not apply under the same conditions of the contact transparencies as the Andreev process which are relevant to different devices. In particular, the authors should remove the third sentence of the abstract.

Now that I have explained my opinion about those issues of communication, I move to the following comments on the physics:

2) It seems to me that there is discrepancy between the claim that the authors make about relevance to experiments (and even suggesting "better relevance" than Andreev processes, see above) and the complete absence of discussion of the experimental consequences of their own proposal on topology. Could the authors deliver a schematic device and protocol that can be used to probe what they say about topology ? Should this experiment on topology be considered as being simple or difficult ? This would be a great help regarding the experimental audience.

3) I am confused by what the authors call "sequential tunneling". Do they mean "sequential tunneling of single-electron states" or "sequential tunneling of Cooper pairs" ? It seems to me that tunneling one Cooper pair at a time should not produce quartet-like signal, but maybe I am wrong on this point. A related question is whether the authors' proposal relies on "breaking Cooper pairs" as in Cooper pair splitting, or whether each Cooper pair remains bound throughout the entire quantum transport process. I could well imagine that Coulomb interaction is fully compatible with breaking two Cooper pairs e.g. (e up 1, e down 1) and (e up 2, e down 2) which could sequentially tunnel in the following order in the different leads: e up 1 / e down 2 / e down 1 / e up 2. Then, this would not be very far from the quartet-Andreev mechanism, wouldn't it ?

4) Also, in this sequential tunneling picture, I would like the authors to tell whether there is an upper bound to the delay time between two tunneling events. Arbitrarily large delay time between two sequential tunneling events would sound like being paradoxical, because I expect quantum coherence to be lost on long time scale. Then, what is the upper bound to the delay time between two tunneling events ? Is it set by the inverse of the superconducting gap or by the inverse of the Coulomb interaction or by something else ?

5) Also, I would like the authors to comment about a simple intuitive picture for entanglement in their proposal. Apparently, nontrivial topology implies quantum effects and, likely, also entanglement. Are quantum correlations and entanglement simple consequences of the number-phase conjugation, as it would be there for two terminals within the model of the authors ? Is entanglement a consequence of the two-component Bogoliubov-de Gennes spinor acting like a spin-1/2, as it is the case for Andreev molecules ? [see references A an B below]. Is entanglement a consequence of four-fermion exchange as in the quartet mechanism ?

6) It seems to me that the authors did not consider the following consequence of their work. As far as I understand, there is the following classification for nonlocality in three-terminal Josephson junctions [made with S1-S2-S3 three-terminal junctions where S1 and S3 are laterally connected on S2 through any type of weak link].

I. Equilibrium + absence of Coulomb interactions: nonlocality is limited by the superconducting coherence length, see the Andreev molecules [References A-E below].

II. Finite bias voltage on the quartet line + absence of Coulomb interactions: There is long range nonlocality due to the interplay between Floquet and quasiparticle propagation [Reference F below], at separations that exceed the superconducting coherence length by orders of magnitude.

It seems to me that the authors could add and discuss or demonstrate the following item for a double Josephson junction:

III. Equilibrium + electrostatics (i.e. the authors' proposal) imply long range nonlocality in S1-S2-S3 at separation between the interfaces that can exceed the superconducting coherence length by orders of magnitude. Only electrostatic coupling between the S1-S2 and the S2-S3 interfaces is needed and no bias voltage is required.

This discussion of the range of the effect is of interest towards experimental audience, in a way that is more practical than topology.

In summary, the paper is interesting. I encourage the authors to take the above comments into account, prepare a revised version and resubmit their work.

[A] J.D. Pillet, V. Benzoni, J. Griesmar, J.-L. Smirr, and \c{C}.\"{O}. Girit, Nonlocal Josephson Effect in Andreev Molecules, Nano Lett. {\bf 19}, 7138 (2019).

[B] J.-D. Pillet, V. Benzoni, J. Griesmar, J.-L. Smirr, and \c{C} \"O Girit, Scattering description of Andreev molecules, SciPost Phys. Core {\bf 2}, 009 (2020).

[C] Z. Scher\"ubl, A. P\'alyl and S. Csonka, Transport signatures of an Andreev molecule in a quantum dot-superconductor-quantum dot setup, Beilstein J. Nanotechnol. {\bf 10}, 363 (2019).

[D] V. Kornich, H.S. Barakov, and Yu.V. Nazarov, Fine energy splitting of overlapping Andreev bound states in multiterminal superconducting nanostructures, Phys. Rev. Research {\bf 1}, 033004 (2019).

[E] V. Kornich, H. S. Barakov and Yu. V. Nazarov, Overlapping Andreev states in semiconducting nanowires: competition of 1D and 3D propagation, Phys. Rev. B {\bf 101}, 195430 (2020).

[F] R. M\'elin, Ultralong-distance quantum correlations in three-terminal Josephson junctions, arXiv:2103.07971, Phys. Rev. B in press.

  • validity: -
  • significance: -
  • originality: -
  • clarity: -
  • formatting: -
  • grammar: -

Author:  André Melo  on 2021-10-07  [id 1819]

(in reply to Report 1 by Regis Melin on 2021-07-22)

We thank Prof. Régis Mélin for the time and effort in reviewing our manuscript. Below we address the technical questions posed in the report.

When reading the abstract and the manuscript, I did not understand whether or not the authors want to deliver the message that Coulomb interactions are at work in the three recent experiments reporting the quartets.

The mechanism we put forward in the manuscript for multiplet supercurrent relies only on local tunneling processes, which occur at arbitrary junction transparencies. We do not claim that this mechanism fully explains existing experiments. Rather we show that it cannot be ruled out as a source (i.e., a partial source) of the main experimental signature of quartets: an enhancement of the conductance at commensurate voltages. We believe the calculations provided in the manuscript give sufficient evidence of this.

In particular, the authors should remove the third sentence of the abstract.

The third sentence in the abstract reads "However, this approach requires precise control over microscopic states and is obscured by dissipative current". The Andreev mechanism for multiplet supercurrent relies on delicate charge transfer processes between multiple terminals, and is therefore sensitive to microscopic details of the scattering regions. Furthermore, SNS are subject to high dissipation. This is evident in the relatively low conductance values reported in experiments with SNS devices, see for example Fig. 2 of 10.1073/pnas.1800044115. We therefore believe the sentence is accurate and have retained it in the resubmitted version of our manuscript.

2) It seems to me that there is discrepancy between the claim that the authors make about relevance to experiments (and even suggesting "better relevance" than Andreev processes, see above) and the complete absence of discussion of the experimental consequences of their own proposal on topology. Could the authors deliver a schematic device and protocol that can be used to probe what they say about topology ? Should this experiment on topology be considered as being simple or difficult ? This would be a great help regarding the experimental audience.

Although our proposal accounts for the effects of Berry curvature of the circuit ground state, there is no topology at play. In fact, the Chern number of the circuit in the 2d manifold of the phases of the leads is zero due to time-reversal symmetry (see 10.1103/PhysRevResearch.3.013288). To probe the geometrical contribution to the multiplet supercurrent, we observe that it scales linearly with the applied voltage, while the adiabatic component is constant. Therefore, the geometric effects can be observed by measuring the current-phase relation at different voltage values (see Figure 4).

3) I am confused by what the authors call "sequential tunneling". Do they mean "sequential tunneling of single-electron states" or "sequential tunneling of Cooper pairs"?

We are referring to sequential tunneling of Cooper pairs. This is the only charge transfer process allowed by the tunneling Hamiltonian $\cos \varphi$.

4) Also, in this sequential tunneling picture, I would like the authors to tell whether there is an upper bound to the delay time between two tunneling events.

We study the circuit in the adibatic limit and hence there is no lower bound on the applied voltage (or equivalently an upper bound on the delay time). Because this is a ground state property, there is no loss of coherence at long time scales.

5) Also, I would like the authors to comment about a simple intuitive picture for entanglement in their proposal.

The only entanglement present in the device is that of the BCS ground state. In other words, the entanglement present in the BCS ground state is sufficient to generate multiplets. The author may also wish to consider our response to Report 2 below, where we draw a connection to quantum dot models.

6) It seems to me that the authors did not consider the following consequence of their work. As far as I understand, there is the following classification for nonlocality in three-terminal Josephson junctions [made with S1-S2-S3 three-terminal junctions where S1 and S3 are laterally connected on S2 through any type of weak link] (...) III. Equilibrium + electrostatics (i.e. the authors' proposal) imply long range nonlocality in S1-S2-S3 at separation between the interfaces that can exceed the superconducting coherence length by orders of magnitude. Only electrostatic coupling between the S1-S2 and the S2-S3 interfaces is needed and no bias voltage is required.

We cannot confirm this observation because the classical device with no Coulomb repulsion also generates multiplet supercurrent. Therefore we do not think our work is sufficient to draw such a conclusion, and would rather avoid speculating on this topic.

---

## Round 3 · Referee Report · Anonymous (Referee 2) · 2021-8-11

Strengths

1) Possibly creating a link between the two communities studying Andreev physics and SIS based quantum circuits. 2) Clearly and pedagogically written.

Weaknesses

1) Rather short, could go into more depth at times.

Report

In this work, the authors revisit a three-terminal circuit based on standard SIS junctions under the new viewpoint of multiplet physics, typically studied in a very different regime of junctions with strong tunnel coupling (where Andreev bound states form). The authors consider in particular the dc current response to a commensurate voltage drive, focussing on the quartet line, which, according to the authors, cannot be distinguished (at least qualitatively) from the one studied in the Andreev regime of SNS junctions.

In spite of its brevity, the paper picks up on an interesting question, which has the potential of linking the community studying Andreev physics with the one investigating quantum circuits based on standard SIS junctions. Their work might also open up a number of follow-up questions, which could ignite further research into the transport physics of multi-terminal junctions. I therefore think, a publication can be recommended. However, it seems to me that in some places of the manuscript, there is some potential to go into more depth. I would kindly ask the authors to make corresponding changes (see requested changes below), and consider in addition the following basic thoughts.

1) It seems to me, that the main point of the authors is that in order to see a quartet line, one only needs is a mechanism which renders the trivial cosine dependence of the individual Josephson junction energies nontrivial, resulting in a total energy (or energy spectrum) which has Fourier components of the form cos(phi1+phi2). Of course, both Andreev physics, or the here considered charging energy term do the trick. This claim may therefore be true, but I think it needs to be worked out a little more explicitly, see subsequent thoughts. Also, as a little bit of a side point, I wonder if there is not an analogy to be made to classical sequential electron tunneling in NIN junctions? Here, a single tunnel junction will emit electrons in an uncorrelated Poissonian fashion, however two junctions in series (creating a charge island or a quantum dot in the middle) will create transport statistics deviating from the Poissonian distribution. It would not surprise me, if there were some existing works on classical multi-terminal transport, where nontrivial cross-correlations emerge, which could be considered as a classical analogy to the here considered effect.

2) The above leads me to wonder, whether it is not all essentially a question of creating an energy gap (either via the orbital interference effect in the Andreev physics, or via the charging energy term considered by the authors)? Because, in order to measure the quartet line along the lines proposed by the authors, it seems that there needs to be an adiabatic regime, requiring a finite energy spacing to avoid Landau-Zener transitions. Note that the EC->0 limit the authors discuss in the beginning, even though educational, does actually not fit the bill. For vanishing EC, the energy spacing will vanish, such that the adiabatic transport regime the authors discuss throughout the paper, cannot be reached. Another way of expressing the same is that I doubt that the system in the EC->0 regime would adiabatically change its state (the value of \varphi) to the time-dependent minimum value as the phases phi1,phi2 progress in time. Perhaps this is a caveat worth mentioning in the paper.

3) Ultimately, I expect the question is, whether one can map both kinds of circuits (multi-terminal SNS and multi-terminal SIS) to simply a Hamiltonian depending on two phases, H(phi1,phi2), which encompasses the current operator via the standard definition I_alpha=dH/dphi_alpha, such that there is no true distinguishing feature in the properties of the Hamiltonian in the most generic case (either with respect to symmetry/topology or other). It is however not clear to me, whether the authors claim that they have already shown the analogy to the most general extent, or whether this belongs on the "to-do-list" for subsequent work?

4) As a final thought, could it be, that the analogy between the two systems becomes increasingly nontrivial if one considers not only current expectation values, but in addition current-current correlations, or higher moments of the transport statistics?

I have the impression that some aspects of the above thoughts are already included in the manuscript, even though sometimes only in between the lines. Perhaps the authors could make such or similar thoughts more explicit.

Requested changes

1) I think the minimisation problem in the limit EC=0 has a very simple analytic solution, where \varphi = arctan(x) and x only depends on the junction energies and phi1,phi2, and the resulting current to ground should be of the form of sin(arctan(x))=x/sqrt{1+x^2}. Perhaps there could be some use in showing this result explicitly, since a subsequent expansion of the current in orders of x would much more explicitly show the appearance of the sought-after multiplet terms.

2) Also, as mentioned in the main report, in the limit EC->0, the system probably lacks a mechanism, which would allow for \varphi to "adjust" its state to the updated minimal value. The authors might want to comment on this caveat.

3) I am not sure 100% if I correctly understand the meaning of the quantity plotted in Fig. 3b, the so-called "maximum over offset phase of the average current". Could the authors please explain this in more detail, or with different words?

4) In particular with regard to the additional geometric contribution the authors discuss at the end, I think that an additonal plot of the actual dc quartet supercurrent (with and without the geometric correction) might help to get a feeling for the predicted effect.

5) The predicted indistinguishability of multiplet signals between SNS and SIS multi-terminal devices is a bold claim, and should be justified more deeply than it currently is. Perhaps, the thoughts laid out in the main report could help convey, where I see a possibility to improve the discussion and interpretation of the main result of this work.

  • validity: high
  • significance: high
  • originality: good
  • clarity: top
  • formatting: excellent
  • grammar: good

Author:  André Melo  on 2021-10-07  [id 1820]

(in reply to Report 2 on 2021-08-11)

We thank the referee for the time and effort in reviewing our manuscript. Below we address the technical questions posed in the report.

Also, as a little bit of a side point, I wonder if there is not an analogy to be made to classical sequential electron tunneling in NIN junctions?

We thank the referee for this remark. This is an interesting question, however we do not have an immediate answer. Given that, we believe it is beyond the scope of our manuscript.

The above leads me to wonder, whether it is not all essentially a question of creating an energy gap (either via the orbital interference effect in the Andreev physics, or via the charging energy term considered by the authors)?

The presence of an energy gap allows the physics we describe to be dissipationless, which would not be true in the NIN case.

Ultimately, I expect the question is, whether one can map both kinds of circuits (multi-terminal SNS and multi-terminal SIS) to simply a Hamiltonian depending on two phases, H(phi1,phi2),

This logic is likely to be true as long as the phase-dependence of the energy of the ground state is a nonlinear function of the two phases. However, we can develop a more direct connection between our SIS circuit and the SNS devices. It is known that the conventional Cooper pair transisor (two terminal SIS with two junctions in series) in the deep charging regime ($E_C \gg E_J$) is exactly mappable to that of a single-level quantum dot with weak coupling strengths $\Gamma \ll \Delta$ to a pair of superconducting reservoirs (see 10.1006/spmi.1996.0116). In this analogy between the SNS and SIS, $\Gamma$ relates to $E_J$, and a level offset energy relates to offset charge. The same analogy extends to the multi-terminal case (see, e.g. 10.1103/PhysRevLett.124.197002). Such dots would exhibit exactly the same kind of quartet supercurrent described in our work. On the other hand, quartet processes that entangle Cooper pairs from different leads require intermediate states with broken Cooper pairs, and thus they would be suppressed by factors of $\Gamma/\Delta$. Many or even all of the Andreev levels of experimental multiterminal and multichannel SNS devices may be weakly coupled to the superconducting reservoirs, and those levels would predominantly contribute the kind of quartet supercurrent described in our manuscript. We have included these considerations in the revised version of the manuscript.

1) I think the minimisation problem in the limit EC=0 has a very simple analytic solution, where \varphi = arctan(x) and x only depends on the junction energies and phi1,phi2, and the resulting current to ground should be of the form of sin(arctan(x))=x/sqrt{1+x^2}. Perhaps there could be some use in showing this result explicitly, since a subsequent expansion of the current in orders of x would much more explicitly show the appearance of the sought-after multiplet terms.

We thank the referee for this remark. We have added an explicit analytical solution for the classical case (see equation 1).

2) Also, as mentioned in the main report, in the limit EC->0, the system probably lacks a mechanism, which would allow for \varphi to "adjust" its state to the updated minimal value. The authors might want to comment on this caveat.

We assume in our discussion that the classical circuit has damping, which justifies treating $\varphi$ adiabatically. We have added an explicit explanation of this in the manuscript.

3) I am not sure 100% if I correctly understand the meaning of the quantity plotted in Fig. 3b, the so-called "maximum over offset phase of the average current". Could the authors please explain this in more detail, or with different words?

We dub this quantity the critical quartet supercurrent: the current $I_0$ maximized over the average phase of the electrodes $\max_{\phi_1 + \phi_2} \langle I^0(\phi_1(0) + Vt, \phi_2(0) - Vt)\rangle$. We have also added this more detailed description to the caption of Fig. 3b.

4) In particular with regard to the additional geometric contribution the authors discuss at the end, I think that an additonal plot of the actual dc quartet supercurrent (with and without the geometric correction) might help to get a feeling for the predicted effect.

We thank the referee for this suggestion. In Fig. 4b we now plot a quartet current-phase relation with and without the geometric contribution.

5) The predicted indistinguishability of multiplet signals between SNS and SIS multi-terminal devices is a bold claim, and should be justified more deeply than it currently is. Perhaps, the thoughts laid out in the main report could help convey, where I see a possibility to improve the discussion and interpretation of the main result of this work.

As stated earlier, there is an exact analogy between weak-coupling dots and the deep-charging CPT. Deviating from this combination of regimes will indeed result in quantitative differences. Our point is that the presence of an enhancement of conductance due to quartet processes is shared by both systems.

---

## Round 4 · Referee Report · Regis Melin (Referee 1) · 2021-10-15

Report

The authors convincingly answered my comments. The manuscript is important to the field and will certainly inspire future works. I am glad to recommend publication.

---

## Round 4 · Referee Report · Anonymous (Referee 2) · 2021-11-3

Report

The authors have responded to all remarks made by all referees. I can wholeheartedly recommend its publication.

---

## Round 4 · List of Changes

• Added analytical expression of the classical quartet supercurrent
  • Computed quartet current-phase relation with and without Berry curvature contribution
  • Clarified quantity plotted in Fig. 3b
  • Extended discussion about connection with SNS experiments

---

## Editorial Decision

published